# Influence of Anthropometric Characteristics on Ice Swimming Performance—The IISA Ice Mile and Ice Km

**DOI:** 10.3390/ijerph18136766

**Published:** 2021-06-24

**Authors:** Beat Knechtle, Ram Barkai, Lee Hill, Pantelis T. Nikolaidis, Thomas Rosemann, Caio Victor Sousa

**Affiliations:** 1Institute of Primary Care, University Hospital Zurich, 8006 Zurich, Switzerland; Thomas.Rosemann@usz.ch; 2Medbase St. Gallen Am Vadianplatz, 9000 St. Gallen, Switzerland; 3International Ice Swimming Association (IISA), 7798 Cape Town, South Africa; ram.iceswimming@gmail.com; 4Department of Pediatrics, McMaster University, Hamilton, ON L8N 3Z5, Canada; hilll14@mcmaster.ca; 5School of Health and Caring Sciences, University of West Attica, 12243 Egaleo, Greece; pademil@hotmail.com; 6Laboratory of Exercise Testing, Hellenic Air Force Academy, 13671 Acharnes, Greece; 7Bouve College of Health Sciences, Northeastern University, Boston, MA 02115, USA; cvsousa89@gmail.com

**Keywords:** body height, body mass, body fat, endurance performance, water sport

## Abstract

Ice swimming following the rules of IISA (International Ice Swimming Association) is a recent sports discipline starting in 2009. Since then, hundreds of athletes have completed an Ice Mile or an Ice Km in water colder than 5 °C. This study aimed to expand our knowledge about swimmers completing an Ice Mile or an Ice Km regarding the influence of anthropometric characteristics (i.e., body mass, body height, and body mass index, BMI) on performance. We analyzed data from 957 swimmers in the Ice Km (590 men and 367 women) and 585 swimmers in the Ice Mile (334 men and 251 women). No differences were found for anthropometric characteristics between swimmers completing an Ice Mile and an Ice Km although water temperatures and wind chill were lower in the Ice Km than in the Ice Mile. Men were faster than women in both the Ice Mile and Ice Km. Swimming speed decreased significantly with increasing age, body mass, and BMI in both women and men in both the Ice Mile and Ice Km. Body height was positively correlated to swimming speed in women in the Ice Km. Air temperature was significantly and negatively related to swimming speed in the Ice Km but not in the Ice Mile. Water temperature was not associated with swimming speed in men in both the Ice Mile and Ice Km but significantly and negatively in women in Ice Km. In summary, swimmers intending to complete an Ice Mile or an Ice Km do not need to have a high body mass and/or a high BMI to swim these distances fast.

## 1. Introduction

Cold water swimming, including ice swimming and winter swimming, has been a popular recreational past time for nearly 500 years [1,2]. Historically, ice swimming is traced back to the early Russian celebrations of the Epiphany [1]. However, in recent years, ice swimming has steadily grown in popularity and has become an internationally recognized competitive discipline [3].

The International Ice Swimming Association (IISA) was created in 2009 to formalize ice swimming events, including developing specific rules and guidelines to have an event or an individual swim recognized [3]. In 2008, the 2.3 km Swim in Lake Zurich [4] at 4.5 °C played a pivotal role in setting the boundaries and rules in IISA and spawned the idea of establishing IISA. The IISA established the Ice Mile in 2009 as the ultimate achievement of swimming in ice-cold water. In 2014, IISA introduced the Ice Km in addition to the Ice Mile. Since its beginning, several athletes officially completed an Ice Mile or an Ice Km [5]. However, in elite open-water swimmers competing in longer race distances, there is a higher overall competitive level of women compared to men [6]. Overall, IISA has now established rules to officially accept a performance as an Ice Mile [7]. It is important to note that the Ice Mile is swum as an individual challenge to complete it, whereas the Ice Km is swum as part of a swimming competition to complete the distance in the shortest possible time. Cold water and ice swimming are already particularly hazardous and are not without risk [8,9].

The competitive aspect may also introduce other elements of anxiety and increased breathing, which may increase the health risk if not adequately trained and acclimatized [2,10]. In cold water swimming, athletes may face both hypothermia and afterdrop (i.e., a continued drop in core temperature after the human body has been removed from the cold environment) with an increased health risk [2,10,11,12,13,14,15]. Case reports [4,10,11,12] and field studies [16,17,18,19] have shown that both a higher body mass and a higher body fat percentage (e.g., thicker skinfold thickness) might be helpful to endure longer in ice-cold water. Although a study with a sample of 103 recreational open-water swimmers consistently swimming without wetsuits throughout the winter months in the San Francisco Bay with a water temperature range from 9.6 °C to 12.6 °C showed the importance of body mass index (BMI) [20], larger samples of athletes are missing.

Few studies demonstrated performance and participation trends in various open-water long-distance swimming events [16,21,22,23,24,25]. It is also interesting to note that there seems to be a closing of the gender gap in longer events, specifically with women producing faster completion times than men in the Triple Crown of Open Water Swimming [24]. The body composition of female open-water swimmers, with different percentage and distribution of fat tissue, shows several advantages with more buoyancy and less drag in aquatic conditions that determine the small difference between males and females [6].

However, it is also noteworthy that men were faster than women in the Ice Mile and Ice Km events [5]. For long-distance open-water swimmers competing in FINA races from 5 km to 25 km, a lower body mass and a lower lean mass index were related to better finishing positions in World Championships [6].

Therefore, the present study aimed to expand our knowledge about swimmers completing an Ice Mile or an Ice Km regarding the influence of anthropometric characteristics on performance. Based on current findings, we hypothesized that more female and male swimmers with a higher BMI would have successfully finished an Ice Mile than an Ice Km. With increasing length of the swim and increasing duration of staying in ice-cold water, the aspect of a higher BMI would become more critical.

## 2. Materials and Methods

### 2.1. Ethics Approval

This study was approved by the Institutional Review Board of Kanton St. Gallen, Switzerland, with a waiver of the requirement for informed consent of the participants, as the study involved the analysis of publicly available data (1 June 2010).

### 2.2. Data Collection

The data used in this research was obtained from the website of IISA [3]. Not all information was available on the IISA website. Nevertheless, at their discretion, the athletes were requested to share some information allowed to be used by IISA for research purposes. Some of the data were acquired at the application to participate in an event (e.g., age and anthropometric characteristics, such as body mass and body height) and some post-event when results were available (e.g., water and air temperature). Data such as body mass and body height are collected before every race at the entry time. The data for ice miles are collected following a successful Ice Mile attempt on request for verification of the attempt by IISA. All data received were de-identified with no information that may link a data series to a specific swimmer. BMI was also divided following the CDC’s guidelines: normal weight ≥18.5 and <25 kg/m^2^; overweight ≥25 to <30 kg/m^2^; obese ≥30 kg/m^2^ [26]. All swimmers provided written consent for their data to be collected and used for research purposes. Regarding required data about water and air temperature, each swimmer has to follow the rules of IISA [27] to provide the correct data in order for an attempt will be accepted as official for the Ice Mile or Ice Km. Participants with missing anthropometric or weather data were excluded from analysis. For both Ice Mile and Ice Km, wind-chill temperature was calculated from air temperature using the Siple formula [28].
T_apparent_(°C) = 33 + (T_air_ − 33) × (0.474 + 0.454√(v) − 0.0454.v).

### 2.3. Statistical Analysis

We applied Kolmogorov–Smirnov and Levene’s to test for normality and homogeneity, respectively. All variables presented a parametric or homogenous distribution. A general linear model with two factors (two-way ANOVA) was used for environmental conditions, age, and BMI, with sex and distance as independent factors. The primary outcome variable is individual performance; thus’ individual average speed was the dependent variable for all ANOVA models. Individual average race speed (m/s) was calculated individually considering the race time and distance. A general linear model adjusted by age (ANCOVA) was applied considering sex and distance as fixed and random factors, respectively. Effect size was estimated using partial eta square (_p_η^2^). Pairwise comparisons were conducted using the least significant difference (LSD) post-hoc technique. Correlation analyses were conducted using Pearson’s correlation coefficient. All statistical analyses were carried out using Statistical Software for the Social Sciences (IBM^®^ SPSS v.25, Chicago, IL, USA).

## 3. Results

The final sample size included 1542 swimmers (924 men and 618 women). Table 1 summarizes the results for water temperature, wind chill, body mass, body height, BMI, and age for both women and men for both distances. Water temperature and wind chill showed a significant distance effect (F = 404.5; *p* = 0.032; and F = 31,197; *p* = 0.004), with lower temperatures in Ice Km. No significant effects for sex or distance were identified (*p* > 0.05) for body mass, body height, BMI, and age (Table 1).

The ANCOVA model adjusted by age showed a significant sex effect (F = 330.4; *p* = 0.027; _p_η^2^ = 0.997) and distance effect (F = 619.3; *p* = 0.019; _p_η^2^ = 0.998). Post-hoc analysis showed that men were faster than women in both distances, and athletes swimming an Ice Km were faster than athletes swimming an Ice Mile (Figure 1).

The model with sex and BMI categories in Ice Km showed significant effect for sex (F = 30.3; *p* < 0.001; _p_η^2^ = 0.044) and BMI (F = 31.9; *p* < 0.001; _p_η^2^ = 0.089). Similarly, for the Ice Km, we found a significant effect for sex (F = 11.1; *p* = 0.001; _p_η^2^ = 0.027) and BMI (F = 16.8; *p* < 0.001; _p_η^2^ = 0.078). Post-hoc analyses showed that men were faster in all BMI categories but the normal weight in Ice Km. Furthermore, subjects in the obese BMI category were slower regardless of their sex (Figure 2).

Association analyses for men showed that performance was negatively correlated with age (Ice Km: r = −0.343; *p* < 0.001; Ice Mile: r = −0.237; *p* < 0.001). Significant and negative correlations were found between performance and body mass (Ice Km: r = −0.105; *p* = 0.036; Ice Mile: r = −0.143; *p* = 0.040) and BMI (Ice Km: r = −0.118; *p* = 0.018; Ice Mile: r = −0.175; *p* = 0.012). Men’s performance in Ice Km showed a weak correlation with air temperature (r = −0.09; *p* = 0.029), but it was not correlated with water temperature, wind chill, and body height (*p* > 0.05) (Figure 3).

Association analyses for women showed that performance was negatively correlated with age (Ice Km: r = −0.592; *p* < 0.001; Ice Mile: r = −0.402; *p* < 0.001). Significant and negative correlations were found between performance and body mass (Ice Km: r = −0.105; *p* = 0.036; Ice Mile: r = −0.143; *p* = 0.040) and BMI (Ice Km: r = −0.501; *p* < 0.001; Ice Mile: r = −0.134; *p* = 0.059). Women’s performance in Ice Km showed a small correlation with body height (r = 0.151; *p* = 0.013), water temperature (r = −0.120; *p* = 0.022), and air temperature (r = −0.142; *p* = 0.006). No other correlations with women’s performance were found (Figure 4).

Water temperature correlated significantly and positively with both air temperature (r = 0.412, *p* < 0.001) and wind chill (r = 0.397, *p* < 0.0001) (wind chill for both Ice Mile and Ice Km, air temperature only for Ice Km).

## 4. Discussion

This study intended to investigate the influence of anthropometric characteristics (i.e., body mass, body height, BMI) on swimming performance in both the Ice Mile and Ice Km. We hypothesized that more female and male swimmers with a higher BMI would successfully finish an Ice Mile than an Ice Km. The most important findings were (i) swimming speed decreased with increasing age, body mass, and BMI in both women and men in both Ice Mile and Ice Km; (ii) body height was positively correlated to swimming speed in women in Ice Km; and (iii) there were no differences for anthropometric characteristics between swimmers completing an Ice Mile and an Ice Km although water temperatures and wind chill were lower in Ice Km compared to Ice Mile. Further important findings were that men were faster than women in both Ice Mile and Ice Km, air temperature was significantly and negatively related to swimming speed in Ice Km but not in Ice Mile, and water temperature was not associated with swimming speed in men in both Ice Mile and Ice Km but in women in Ice Km.

An important finding was that swimming speed has a small but significant negative correlation to body mass and BMI in both women and men in both Ice Mile and Ice Km. As a result, swimmers with a higher BMI were not faster than swimmers with a lower BMI. These results do not confirm our hypothesis. Several case studies showed that swimming an Ice Mile [4,10,11,12] was completed by a swimmer with a high BMI and/or thick skinfolds. Another case report showed that a swimmer with a BMI of 35.7 kg/m^2^ could swim for 6 h in the water of 9.9 °C without suffering from hypothermia [16]. An early study about Channel swimmers from 1955 reported that swimmers from the race in 1951 were frankly obese [29].

However, a study investigating 103 recreational open-water swimmers swimming during winter months regularly and without wetsuits in the San Francisco Bay at water temperatures from 9.6 °C to 12.6 °C showed that the average BMI of those cold-water swimmers (25.9 kg/m^2^) was lower than controls (i.e., general population) but not different to the BMI of North American masters pool swimmers or international masters pool swimmers [20]. Furthermore, a study investigating elite-level open-water swimmers showed that these swimmers were smaller and lighter than competitive pool swimmers [30].

Based on the present findings, we must assume that a high BMI is not needed for a fast time in both an Ice Mile and an Ice Km for both women and men. A higher body fat might be protective against environmental cold, especially abdominal fat. However, in adults attempting to reach the shore following an accidental immersion in cold water at around 14 °C, abdominal skin fold and percent body fat showed no significant correlation to performance, where only triceps skinfold thickness was a strong predictor of the swimming distance covered [31]. In addition, it has been suggested that an increased body mass might decrease acceleration in swimmers [32] considering concomitant augmentation of swimmers’ surface area [33]. However, the frontal area of the swimmer does not directly affect the drag, especially at low swimming speeds, which is typical for ice swimming. The effective frontal area, i.e., the frontal area plus the trunk incline increases the frontal area [34,35]. Future studies could potentially focus on the optimal body fat percentage for cold water swimming.

A further important finding was that body height was associated with swimming speed in women in the Ice Km where taller women were swimming faster. Little is known regarding the aspect of body dimensions, such as body height, for open-water swimmers. A study investigating the relationship between anthropometric and training characteristics to performance in a 26.4 km open-water swimming event showed that body height, BMI, length of the arm, and swimming speed during training were associated with race time in men. Further, this study showed that swimming speed during training was associated with race performance in women [36]. A study investigating a potential correlation between body dimensions and swimming performance in a 12-h swim showed no association between selected anthropometric characteristics (i.e., body mass, body height, BMI, circumferences of extremities, length of arms and legs, skeletal muscle mass, and fat mass) and performance [37]. 

In open-water swimming in FINA races from 5 km to 25 km, the body composition of female athletes with a different percentage and a different distribution of fat tissue compared to male athletes showed several advantages, with more buoyancy and less drag in aquatic conditions that determine the small difference between males and females in these distances [6]. Most probably, body dimensions are not highly relevant for a fast time in both an Ice Mile and an Ice Km. For other swimmers, such as sprint swimmers, body composition is relevant for performance [38]. Future studies might investigate pre-event preparation (i.e., training and specific event preparation) in ice swimmers.

A notable finding was that female performance was significantly and negatively correlated to water temperature in Ice Km. In other terms, lower water temperatures impaired female performance. No influence of water temperature on swimming performance was found for men. In a first analysis of an Ice Mile and Ice Km with 113 men and 38 women in the Ice Mile and 26 men and 13 women in the Ice Km, water temperature showed no correlation to swimming speed in both events and for both sexes [5]. Interestingly, this result contrasts with a recent review that suggested that women would be faster in colder water due to various anthropometric characteristics, including body fat deposition [25].The advantages of specific anthropometric characteristics might be mitigated due to the shorter swimming duration in cold water.

A last important finding was that air temperature was significantly and negatively related to swimming speed in Ice Km for both women and men, where a higher air temperature was associated with a faster swimming speed. This finding is difficult to explain since wind chill temperature was available for both Ice Mile and Ice Km but air temperature only for Ice Km. One might assume that a higher ambient air temperature could prevent the cooling of the body during swimming. However, this explanation might be highly speculative since the water temperature was highly significantly correlated with air temperature (Ice Km) and wind chill (Ice Mile and Ice Km).

This study has some limitations. Body height and body mass were self-reported and not objectively measured before a swim. A disagreement between self-reported and measured anthropometric data has been reported for athletes [39,40], leading to significant differences between estimated and measured BMI. Secondly, some of the data relied on self-reported performance and, as such, could have influenced results. A further limitation is that we have used BMI as anthropometric characteristics, not fat mass and/or muscle mass. BMI does not separate percentages of fat and muscle regarding body composition. In sprint swimmers, specific anthropometric characteristics (e.g., protein-fat index, index of body composition, percent of skeletal muscle mass, free fat mass, fat mass index, and percent of body fat) are related to swimming performance [38]. A higher BMI is due to more fat mass, as it may also be due to more muscle mass, and this can influence performance. In addition, external conditions, such as wind, currents (lake or river), veering, and trajectory can influence the total time performance in open-water events [41]. Moreover, following IISA regulations, if the swim course is a pool course, the open turns (and 5 m underwater) are allowed. This strongly affects the time performance depending on the events. Furthermore, although an ice-swimming event attempt may not take advantage of a known current as per IISA rules, the current cannot be totally controlled in the outdoor environment and could affect the time performance.

## 5. Conclusions

Swimmers intending to complete an Ice Mile or an Ice Km do not need to have high body mass and/or a high BMI (i.e., increased body fat) to swim these distances fast. The general idea that there is an optimal body composition associated with success in ice swimming do seem to confirm our current results. Training and experience in these swimmers might be of higher importance. Future studies might investigate pre-event preparation (i.e., training and specific event preparation) in ice swimmers. Also, physiological aspects, such as maximum oxygen uptake and swimming economy, might be of interest.

## Figures and Tables

**Figure 1 ijerph-18-06766-f001:**
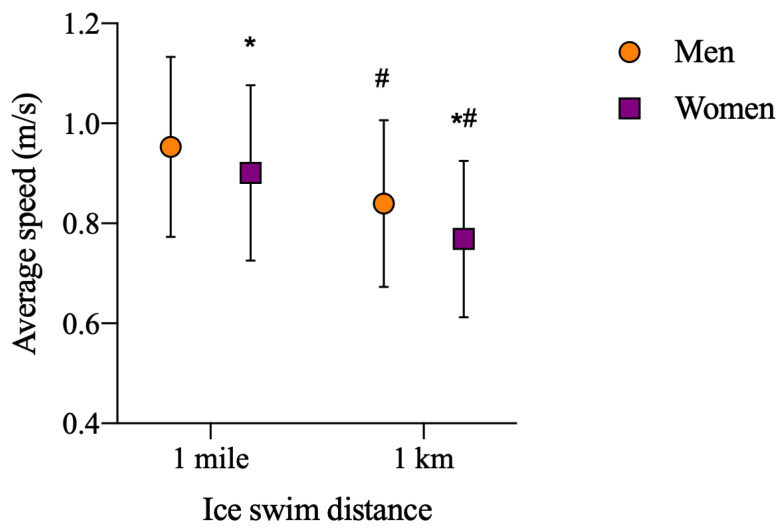
Ice Mile and Ice Km swimming performance (average swimming speed) by sex and distance. *: significant difference between sex. #: significant difference between distance.

**Figure 2 ijerph-18-06766-f002:**
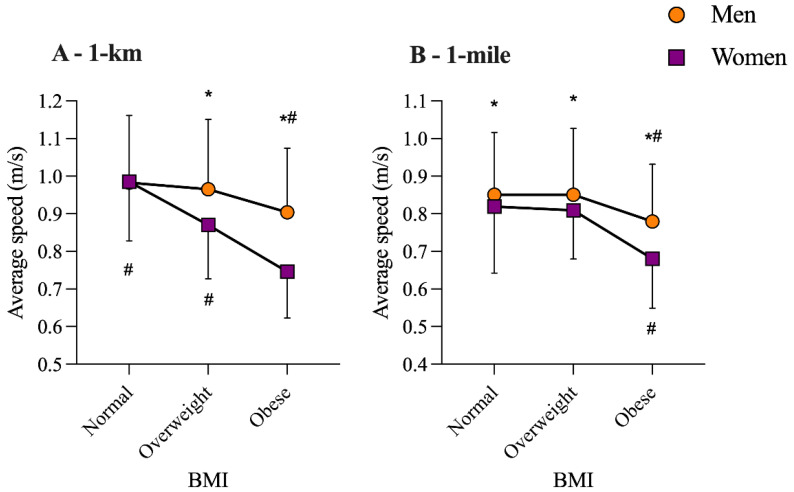
BMI and swimming performance (average speed) by sex of the Ice Mile and Ice Km. *: significant difference between sex. #: significant difference between distance.

**Figure 3 ijerph-18-06766-f003:**
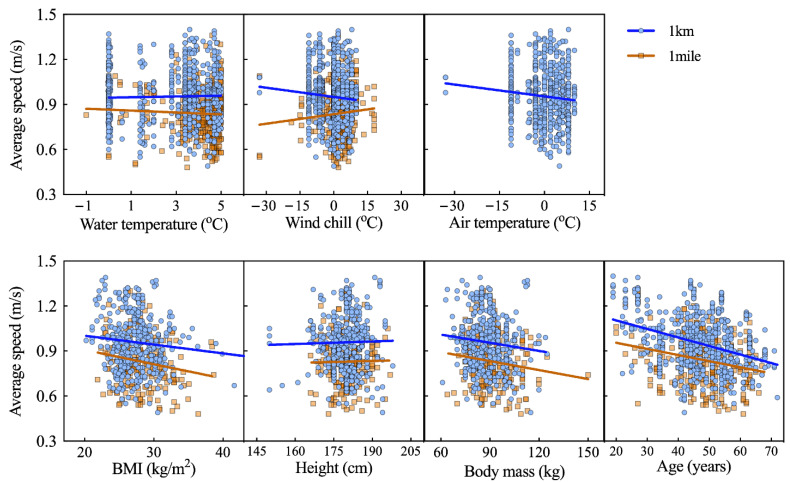
Correlations of men’s performance in Ice Mile and Ice Km (average swimming speed) with age, BMI, body height, body mass, water temperature, and wind chill.

**Figure 4 ijerph-18-06766-f004:**
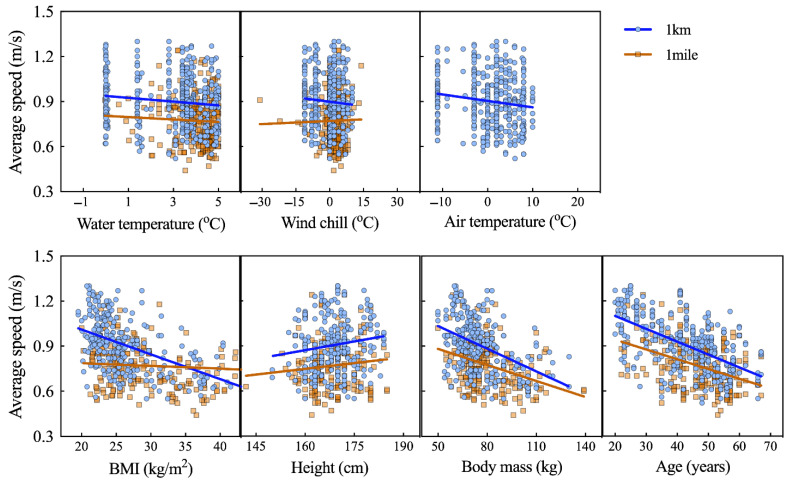
Correlations of women’s performance in Ice Mile and Ice Km (average swimming speed) with age, BMI, body height, body mass, water temperature, and wind chill.

**Table 1 ijerph-18-06766-t001:** Environment conditions, BMI, and age of athletes swimming Ice Km and Ice Mile.

	Ice Km		Ice Mile	
	Men (*n* = 590)	Women (*n* = 367)	Men (*n* = 334)	Women (*n* = 251)
Water temperature (°C)	2.79 ± 1.77	2.91 ± 1.68	4.01 ± 1.16	4.02 ± 0.92
Wind chill (°C)	−0.97 ± 6.09	−1.05 ± 5.70	2.17 ± 6.20	2.06 ± 4.57
Body mass (kg)	88.3 ± 10.5	75.2 ± 15.0	91.8 ± 12.2	81.0 ± 15.6
Body height (cm)	179.6 ± 6.7	169.2 ± 6.7	180.4 ± 6.0	167.9 ± 9.9
BMI (kg/m^2^)	27.4 ± 3.6	26.3 ± 5.3	28.2 ± 3.1	29.4 ± 11.5
Age (years)	46.6 ± 10.8	43.2 ± 12.1	48.3 ± 9.7	46.6 ± 9.7

BMI: body mass index.

## Data Availability

All data are available by the corresponding author upon reasonable request.

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
