# Peer review of "Influence of Anthropometric Characteristics on Ice Swimming Performance—The IISA Ice Mile and Ice Km"

_ijerph, 2021, doi:10.3390/ijerph18136766_

Round 1

Reviewer 1 Report

The authors have responded satisfactorily to the comments.

Author Response

Reviewer 1

The authors have responded satisfactorily to the comments.

Answer: no further changes are required

Reviewer 2 Report

All my concerns have been resolved.

Author Response

Reviewer 2

All my concerns have been resolved.

Answer: no further changes are required

Reviewer 3 Report

ijerph-1269556-peer-review-v1

In the following, I will write in response to the rebuttal letter provided by the authors. I will copy my original suggestions, each answer provided and my new assessment in bold typeface.

L19: Km should be read as km because it is a SI unit.

Answer: We agree with the expert reviewer, however, this term ‘Ice Km’ describes the

specific distance of 1 km swimming in water of colder than 5 °C.

Thank you for this answer. However, regardless of the meaning of the composed noun, a kilometer is written as “km” and therefore I insist on that it should be called “Ice km”. Out of curiosity, I searched on the internet which way was right and I found only one website referring to this term: https://internationaliceswimming.com. In the entire website, never this modality is called “Ice Km”, but ICE KM, which makes it unlikely that the real name was “Ice Km”. 

L17-32: The authors are constantly repeating the term `Ice Mile` and `Ice Km`. Please write

the abstract paying attention to that issue.

Answer: We agree with the expert reviewer, however, this term ‘Ice Km’ describes the

specific distance of 1 km swimming in water of colder than 5 °C. The term ‘Ice Mile’

describes the specific distance of 1 mile swimming in water of colder than 5 °C.

The authors are not considering my suggestion nor are they making logical arguments.

L41-66: In an introductory paragraph, authors must be very careful about space and therefore, they must go straight to the problem they wanted to solve. In my opinion, from lines 41 to 66, too much basic information about the sport which can be accessed elsewhere is presented. Please, reduce as much as possible and give adequate references for the interested reader.

Answer: We agree with the expert reviewer and shorted where appropriate.

In my first round of review, I asked for a reduction between lines 41 and 66. The present version stills shows similar text between lines 41 and 66, so no reduction has been made. Please reduce significantly the amount of information and lines to express it, please.

L92: Please give IRB number. Also, state if the study was performed according to the

Helsinki protocol.

Answer: We agree with the expert reviewer and inserted as suggested.

An IRB number is the Institutional Review Board, usually 4- or 5-digits, possibly followed by a letter. The authors have failed to provide such a number.

L100: It is not clear how anthropometric measures were performed. What kind of instruments 

did you use? Did an anthropometrist carry out measures? If so, was she or he accredited by

ISAAK?

Answer: We agree with the expert reviewer and changed to ‘Some of the data was acquired 

at the application to participate in an event (e.g., age and anthropometric characteristics 

such as body mass and body height) and some post-event when results are available (e.g., 

water and air temperature)’.

So, the source of information is that given by participants when applying for the event? How can you ensure that these values are accurate? I think the minimum requirement for a study like this is that the authors have brought with them some sort of stadiometer and a calibrated scale in the competition. Regarding water and air temperature, a similar flaw is present. What happened when results were not available? It is unclear if data from Table 1 is accurate, and therefore, the rest of the study.

L137: What is a post-hoc analysis? Why did you not perform this analysis in line 130?

Answer: Post-hoc analysis should only be used when an independent factor shows a

significant effect over the dependent variable. Then, the post-hoc analysis can identify 

pairwise differences. The technique used for post-hoc was added to the statistical section.

The authors failed to answer the question. Why did you not perform this analysis in line 130? (between water temperature and wind chill over distance effect). 

L154: Which measure of correlation did you use? Was it Pearson’s? If so, you didn’t inform 

the reader about normality check results.

Answer: We did use Pearson’s correlation coefficient. All data was tested for normality and 

homogeneity. This information is on the statistical analysis section under Methods.

Yes, but you didn’t mention the results of the KS test or at least, that the sample followed a gaussian distribution, so we don’t know if Pearson is correctly used or not.

L278: Most references are not appropriate. Refs 3, 6, 25, 26, 27 are web pages: I think you 

should search for another source of information. Then, more than half the cites are autocites: 

Refs 2, 4, 5, 9, 10, 11, 15, 20, 21, 22, 23, 24, 31, 32, 34, 35

Answer: Some important information about ice swimming is only available from websites. 

With new references the percentage of self-citation has been reduced.

Ok, but reduce the number of self-cites to less than 13%. In a study from Nature: https://www.nature.com/articles/d41586-019-02479-7, the median self-citation rate is 12.7% 

Author Response

Reviewer 3

ijerph-1269556-peer-review-v1

In the following, I will write in response to the rebuttal letter provided by the authors. I will copy my original suggestions, each answer provided and my new assessment in bold typeface.

L19: Km should be read as km because it is a SI unit.

Answer: We agree with the expert reviewer, however, this term ‘Ice Km’ describes the

specific distance of 1 km swimming in water of colder than 5 °C.

Thank you for this answer. However, regardless of the meaning of the composed noun, a kilometer is written as “km” and therefore I insist on that it should be called “Ice km”. Out of curiosity, I searched on the internet which way was right and I found only one website referring to this term: https://internationaliceswimming.com. In the entire website, never this modality is called “Ice Km”, but ICE KM, which makes it unlikely that the real name was “Ice Km”. 

Answer: no

L17-32: The authors are constantly repeating the term `Ice Mile` and `Ice Km`. Please write

the abstract paying attention to that issue.

Answer: We agree with the expert reviewer, however, this term ‘Ice Km’ describes the

specific distance of 1 km swimming in water of colder than 5 °C. The term ‘Ice Mile’

describes the specific distance of 1 mile swimming in water of colder than 5 °C.

The authors are not considering my suggestion nor are they making logical arguments.

Answer: we write throughout the manuscript ‘Ice km’

L41-66: In an introductory paragraph, authors must be very careful about space and therefore, they must go straight to the problem they wanted to solve. In my opinion, from lines 41 to 66, too much basic information about the sport which can be accessed elsewhere is presented. Please, reduce as much as possible and give adequate references for the interested reader.

Answer: We agree with the expert reviewer and shorted where appropriate.

In my first round of review, I asked for a reduction between lines 41 and 66. The present version stills shows similar text between lines 41 and 66, so no reduction has been made. Please reduce significantly the amount of information and lines to express it, please.

Answer: We agree with the expert reviewer and shorted again where appropriate.

L92: Please give IRB number. Also, state if the study was performed according to the

Helsinki protocol.

Answer: We agree with the expert reviewer and inserted as suggested.

An IRB number is the Institutional Review Board, usually 4- or 5-digits, possibly followed by a letter. The authors have failed to provide such a number.

Answer: 01-06-2010; June 1st 2010 from our ethical committee

L100: It is not clear how anthropometric measures were performed. What kind of instruments 

did you use? Did an anthropometrist carry out measures? If so, was she or he accredited by

ISAAK?

Answer: We agree with the expert reviewer and changed to ‘Some of the data was acquired 

at the application to participate in an event (e.g., age and anthropometric characteristics 

such as body mass and body height) and some post-event when results are available (e.g., 

water and air temperature)’.

So, the source of information is that given by participants when applying for the event? How can you ensure that these values are accurate? I think the minimum requirement for a study like this is that the authors have brought with them some sort of stadiometer and a calibrated scale in the competition. Regarding water and air temperature, a similar flaw is present. What happened when results were not available? It is unclear if data from Table 1 is accurate, and therefore, the rest of the study.

Answer: Events like these usually involves only a few athletes, if not a single athlete, so the support crew can give individual and meaningful support if needed. Thus, a study with multiple athletes involves multiple events. We agree that, in-loco measurement would be more accurate, but it could also make the study inviable. Nevertheless, this limitation of self-reported measures is described in the limitations section of the study L260-L265.

Participants with any missing data were excluded from analysis. This information was added to L113-L114.

Data from Table 1 is accurate to the methods used to collect the data. The protocol of self-report has its limitations, as any other scientific method, and the limitations are fully disclosed on the limitations section of the article, as stated above.

L137: What is a post-hoc analysis? Why did you not perform this analysis in line 130?

Answer: Post-hoc analysis should only be used when an independent factor shows a

significant effect over the dependent variable. Then, the post-hoc analysis can identify 

pairwise differences. The technique used for post-hoc was added to the statistical section.

The authors failed to answer the question. Why did you not perform this analysis in line 130? (between water temperature and wind chill over distance effect). 

Answer: Post-hoc analysis is commonly used in ANOVA models to identify specific differences in a model with independent factors with three or more levels. The results that the reviewer is referring to is a 2x2 ANOVA model (Sex and Distance as independent factors), and a significant effect for any factor do not require a post-hoc unless both factors or interaction were significant.

A post-hoc analysis is redundant in the referred case; it only shows the result previously shown with the main effect. 

L154: Which measure of correlation did you use? Was it Pearson’s? If so, you didn’t inform 

the reader about normality check results.

Answer: We did use Pearson’s correlation coefficient. All data was tested for normality and 

homogeneity. This information is on the statistical analysis section under Methods.

Yes, but you didn’t mention the results of the KS test or at least, that the sample followed a gaussian distribution, so we don’t know if Pearson is correctly used or not.

Answer: A new sentence was added to the Statistical analysis section, confirming that the data presented parametric or homogenous distribution.

L278: Most references are not appropriate. Refs 3, 6, 25, 26, 27 are web pages: I think you 

should search for another source of information. Then, more than half the cites are autocites: 

Refs 2, 4, 5, 9, 10, 11, 15, 20, 21, 22, 23, 24, 31, 32, 34, 35

Answer: Some important information about ice swimming is only available from websites. 

With new references the percentage of self-citation has been reduced.

Ok, but reduce the number of self-cites to less than 13%. In a study from Nature: https://www.nature.com/articles/d41586-019-02479-7, the median self-citation rate is 12.7% 

Answer: MDPI has its specific rules regarding self-citation, see https://www.mdpi.com/journal/information/instructions and ‘Authors should not engage in excessive self-citation of their own work’

This manuscript is a resubmission of an earlier submission. The following is a list of the peer review reports and author responses from that submission.

Round 1

Reviewer 1 Report

The authors have analyzed the influence of body height, body weight and BMI on swimming speed in "Ice Mile" and "Ice Km". They found no significant differences.

The main criticism of the work is that the authors consider BMI as an indicator of obesity and fat mass. BMI is a nonspecific index that does not separate the effect of fat mass from muscle mass on total body mass.

To control the effect of adiposity on ice swimming, authors should have used other measures: percentage of fat mass, waist/height ratio, relative fat mass...

As they have not measured, in the discussion of the study, the authors need to justify this. Authors should also comment on the influence of body fat mass and body muscle mass on BMI and swimming performance. Thus, they should write as a limitation of their work that they cannot ensure that a higher BMI is due to more fat mass, as it may also be due to more muscle mass, and this can influence performance.

Line 112. They must replace "body composition" with "BMI". In the work they have not analyzed body composition (percentage of fat weight or percentage of muscle weight), only the weight/height2 ratio. 

It is not correct to talk about body composition when using BMI instead of percentages of fat and muscle. Since the BMI does not separate those components.

Line 125: The final sample size included 1,489 swimmers (924 men and 618 women). The total doesn't match, it's 1,542.).

Reviewer 2 Report

Thank you for presenting an interesting manuscript. It was a pleasure reviewing your work. Obviously this team has put a large amount of work into this research and the crafting of the manuscript. Their expertise on this topic is strong. The manuscript is well organized and the funnel of information is nicely crafted. There is a solid description of how the research was conducted. The results are well presented. However, I have three major concerns about this manuscript:

- some relevant literature on the topic is missing.

- despite long discussed in the manuscript, the same author has already investigated the temperature effect (air and water) and gender effect on ice swimming performance. The novelty is poor.

- The time performance analysis on ice swimming events is a critical point.

Please see my comments/suggestions/explanations below.

- line 18: - line 18: replace “young” with “recent”.

- line 53: In elite open water swimmer there is a higher overall competitive level of women compared to men (Baldassarre, R., Bonifazi, M., Zamparo, P., & Piacentini, M. F. (2017). Characteristics and challenges of open-water swimming performance: a review. International Journal of Sports Physiology and Performance, 12(10), 1275-1284). The authors should point out this inconsistency here.

- line 56: replace “swimming costume” with “swimsuit”. Moreover, what does “standard” mean? What section of the body does a standard costume cover?

- line 61-63: “Paul Georgescu, from Romain, currently performed the longest ice swim in water temperature of 4.43 °C and a distance of 3.5 km in a time of 57:56 min:s [3].” I advise the authors not to include this sentence. In a recent sport such as ice swimming, individual performance is not representative of a performance limit.

- line 65: In the references, I did not find evidence of a higher stroke and kick rate in ice swimming. The authors should clarify.

- line 76-79: This sentence departs from the logical flow of the introduction. Ice swimming competitions are not open water long-distance events. Performance factors change when you swim 10 or 25 km versus a mile or a km in cold water. Furthermore, the anthropometric characteristics of FINA competitors in similar distance events to ice swimming are not mentioned in the introduction.

- Shaw, G., & Mujika, I. (2018). Anthropometric profiles of elite Open-Water swimmers. International journal of sports physiology and performance, 13(1), 115-118.

- Baldassarre, R., Bonifazi, M., Zamparo, P., & Piacentini, M. F. (2017). Characteristics and challenges of open-water swimming performance: a review. International Journal of Sports Physiology and Performance, 12(10), 1275-1284.

- Check the spacing between words in the manuscript.

- line 96: more information about the range of ice swimming dates and events analyzed should be provided.

- line 117: the external conditions in open water events, such as wind, currents (lake or river), veering and trajectory, could influence the total time performance (Saavedra, J. M., Einarsson, I., Sekulic, D., & Garcia-Hermoso, A. (2018). Analysis of pacing strategies in 10 km open water swimming in international events. Kinesiology, 50(2.), 243-250). Moreover, following IISA regulations, if the swim course is a pool course the open turns (and 5m underwater) are allowed. This strongly affects the time performance depending on the events. Furthermore, although an Ice Swimming event attempt may not take advantage of a known current as per IISA rules, the current cannot be totally controlled in the outdoor environment and could affect the time performance. Previous papers for FINA-regulated open water events use ranking as the primary outcome variable and not time performance. For ice swimming this is not possible. I advise the authors to highlight this as a strong limitation of this analysis.

- line 125-129: these results are already presented in table 1.

- line 136-137: clarify which dependent variable was investigated using ANCOVA.

- line 137: describe in the statistical section which technique for post-hoc analysis was used.

- line 177: “body mass” or “body weight”? authors should standardize throughout the manuscript.

- line 184: “(iii) men were faster than women in both 'Ice Mile' and 'Ice Km'”, “(v) air 185 temperature was significantly and negatively related to swimming speed in 'Ice Km' but not in 'Ice Mile'”, “(vi) water temperature was not associated with swimming speed in men in both 'Ice Mile' and 'Ice Km' but in women in 'Ice Km'”. These findings were not the aims of the present research. The results are already previously highlighted by the same author and therefore poor in novelty here. (Knechtle, B., Rosemann, T., & Rüst, C. A. (2015). Ice swimming–‘Ice Mile’and ‘1 km Ice event’. BMC Sports Science, Medicine and Rehabilitation, 7(1), 1-7.). Authors should therefore moderate the discussion of this result.

- line 203-204: “…more energy is expended to move the body through water” I don't find this in Wallingford. Authors should clarify scientific support or otherwise justify.

- line 205-207: This justification is not complete. The frontal area of the swimmer does not directly affect the drag, especially at low speeds typical of ice swimming. The author must consider the effective frontal area, i.e. the frontal area plus the trunk incline (this implies an increase in frontal area).

- Gatta, G., Cortesi, M., Fantozzi, S., & Zamparo, P. (2015). Planimetric frontal area in the four swimming strokes: Implications for drag, energetics and speed. Human movement science, 39, 41-54.

- Zamparo, P., Gatta, G., Pendergast, D., & Capelli, C. (2009). Active and passive drag: the role of trunk incline. European journal of applied physiology, 106(2), 195-205.

- line 216-230: authors should complete discussions with the findings of these papers:

- Baldassarre, R., Bonifazi, M., Zamparo, P., & Piacentini, M. F. (2017). Characteristics and Challenges of Open-Water Swimming Performance: A Review. International Journal of Sports Physiology and Performance, 12(10), 1275–1284. doi:10.1123/ijspp.2017-0230

- Pugh LG, Edholm OG. The physiology of channel swimmers. Lancet (London, England). 1955;269(6893):761-768.

- VanHeest JL, Mahoney CE, Herr L. Characteristics of Elite Open-Water Swimmers. J Strength Cond Res. 2004;18(2):302.

Reviewer 3 Report

L19: Km should be read as km because it is a SI unit.

L17-32: The authors are constantly repeating the term `Ice Mile` and `Ice Km`. Please write the abstract paying attention to that issue.

L32: BMI measures fat but also muscle and bones. The sentence is misleading.

L41: I don’t think IISA is a trademark registered company (R) , but that their intellectual property is copyrighted (C). Even so, my opinion is that neither of them are necessary for a scientific paper.

L46: a unified must be read as an unified

L49: Ice Mile is sticked to in

L53: In line 47, the author stated 4.5 ºC and now +5.0 ºC. Please, be consistent with sign

L54: meters should be read as m

L61: Space needed between sentences

L61: What is Romain? Please define

L63: See L61

L64: See L61

L69: There are plenty of mistakes like this one (fail to leave a space between sentences, after comma, etc). I will no longer advise against this, but authors must be very careful in correcting these basic issues.

L41-66: In an introductory paragraph, authors must be very careful about space and therefore, they must go straight to the problem they wanted to solve. In my opinion, from lines 41 to 66, too much basic information about the sport which can be accessed elsewhere is presented. Please, reduce as much as possible and give adequate references for the interested reader.

L76-83: The problem statement is not fully developed. The authors failed to point out the research already conducted and how your hypothesis may give light to the problem.

L92: Please give IRB number. Also, state if the study was performed according to the Helsinki protocol.

L100: It is not clear how anthropometric measures were performed. What kind of instruments did you use? Did an anthropometrist carry out measures? If so, was she or he accredited by ISAAK?

L118: meters/second should be read as m/s

L126: 4 should be read as 4.0

L127: same with 5

L130: Remove apostrophes from distance, sex and even from Ice Mile and the rest

L134: Body weight should be read as Body mass because weight is a force measured in N

L137: What is eta squared? Please state

L137: What is a post-hoc analysis? Why did you not performed this analysis in line 130?

L137: A post hoc analysis is not intended to show if men were faster than women. A mere descriptive analysis would have sufficed. Why did you perform this analysis?

L147: See L137

L154: Which measure of correlation did you use? Was it Pearson’s? If so, you didn’t inform the reader about normality check results.

L180-188: I don’t think author can claim such relationships with the following r values for men: r = -0.343, r=-0.105, r=-0.143, r=-0.118, r =-0.09 and for women r = -0.105, r = -0.143, r = -0.134, r = 0.151, r = -0.120.

The variances explained by these r values are computed as r^2 and for the claimed r values, these are as low as: 

Men

r = -0.343, → 11.5%

r=-0.105, → 1.1%

r=-0.143, → 2%

r=-0.118, → 1.4%

r =-0.09 → 0.8%

women 

r = -0.105, → 1.1%

r = -0.143, → 2%

r = -0.134, → 1.8%

r = 0.151, → 2.2%

r = -0.120. → 1.4%

The only meaningful result is the association analyses for women: r = -0.592 and r = -0.501 and stil, the practical application is very limited: 32% and 25% of variance explained. Therefore, the results of these study is that there is no influence of anthropometric characteristics (i.e., body mass, body height, BMI) on swimming performance in both 'Ice Mile' and 'Ice Km'.

L169: What is the point in stating that water, wind, and air temperatures correlate with each other? Is it possible that water temperature increases whereas air temperature decreases? I don’t think so. I am not talking about different temperatures in both mediums, but a tendency (association) is very rare. Even so, how is it relevant to the study?

L278: Most references are not appropriate. Refs 3, 6, 25, 26, 27 are web pages: I think you should search for another source of information. Then, more than half the cites are autocites: Refs 2, 4, 5, 9, 10, 11, 15, 20, 21, 22, 23, 24, 31, 32, 34, 35